# Synthesis, Characterization, and Applications of Silver Nano Fibers in Humidity, Ammonia, and Temperature Sensing

**DOI:** 10.3390/mi12060682

**Published:** 2021-06-10

**Authors:** Haroon-Ur Rashid, Muhammad Ali, Mahidur R. Sarker, Sawal Hamid Md Ali, Naseem Akhtar, Nadir Ali Khan, Muhammad Asif, Sahar Shah

**Affiliations:** 1Department of Electronics, University of Peshawar, Peshawar 25120, Pakistan; ali.muhammad@uop.edu.pk (M.A.); nakhan@uop.edu.pk (N.A.K.); m.asif@uop.edu.pk (M.A.); 2Materials Research Laboratory, Department of Physics, University of Peshawar, Peshawar 25120, Pakistan; 3Institute of IR 4.0, Unverisity Kebangsaan Malaysia, Bangi 43600, Malaysia; 4Department of Electrical, Electronic and Systems Engineering, Faculty of Engineering and Built Environment, University Kebangsaan Malaysia, Bangi 43600, Malaysia; sawal@ukm.edu.my; 5Department of Chemistry, Bacha Khan University Charsadda, Charsadda 24420, Pakistan; nasimakhtarchd@gmail.com; 6School of Physics and Electronics, Central South University, Changsha 410083, China; sahar-shah@ele.qau.edu.pk

**Keywords:** silver nanoparticles, nanofibers, humidity, ammonia, temperature sensor

## Abstract

The promising chemical, mechanical, and electrical properties of silver from nano scale to bulk level make it useful to be used in a variety of applications in the biomedical and electronic fields. Recently, several methods have been proposed and applied for the small-scale and mass production of silver in the form of nanoparticles, nanowires, and nanofibers. In this research, we have proposed a novel method for the fabrication of silver nano fibers (AgNFs) that is environmentally friendly and can be easily deployed for large-scale production. Moreover, the proposed technique is easy for device fabrication in different applications. To validate the properties, the synthesized silver nanofibers have been examined through Fourier transform infrared spectroscopy (FTIR), scanning electron microscopy (SEM), and X-ray diffraction (XRD). Further, the synthesized silver nanofibers have been deposited over sensors for Relative humidity (RH), Ammonia (NH_3_), and temperature sensing applications. The sensor was of a resistive type, and found 4.3 kΩ for relative humidity (RH %) 30–90%, 400 kΩ for NH_3_ (40,000 ppm), and 5 MΩ for temperature sensing (69 °C). The durability and speed of the sensor verified through repetitive, response, and recovery tests of the sensor in a humidity and gas chamber. It was observed that the sensor took 13 s to respond, 27 s to measure the maximum value, and took 33 s to regain its minimum value. Furthermore, it was observed that at lower frequencies and higher concentration of NH_3_, the response of the device was excellent. Furthermore, the device has linear and repetitive responses, is cost-effective, and is easy to fabricate.

## 1. Introduction

Nowadays, nanoparticles (NPs) are gaining great interest because of their nanostructure, small size, large surface area, high thermal stability, and excellent physical and chemical properties, which cannot be achieved by bulk materials [1]. A small size (ranging from 1 nm to 100 nm) [2], one dimension [3], and a large surface area allow these NPs to attach to biological entities without changing their properties and function. Some materials behave differently at the nanoscale, and they emerge with new characteristics, like aluminum, as they become explosive or their melting point changes. Similarly, AgNPs have revealed new properties and act as antibacterial agents [4]. However, recent studies on NPs and nanofibers showcase the potential risks of NP aerosol release, and allow amore balanced benefit/risk analysis. For example, many studies highlight NP emissions due to coatings, paints [5], and tiles [6]. Cases of NPs exposure in the field of occupational hygiene at coating workplaces have been reported [7]. A small size and a large surface area permit NPs to make strong bonds with surfactant molecules. Among these physical and chemical applications of NPs, silver nanoparticles (AgNPs) have become the main focusing area for the researchers in the field of nanotechnology [8]. AgNPs can be used in anti-bacterial agents, medical applications, food, and healthcare. AgNPs can also be used in anti-bacterial coatings, wound dressings found in socks, refrigerators, and sports clothing because they have the ability to kill germs [9].AgNPs can also be used in liquid form for coatings and sprays within a shampoo [10]. Similarly, AgNPs in the form of silver nanofibers (AgNFs) can be employed in water purification systems as a filtration membrane [11] that stores silver ions, forming a thin membrane layer that creates a protective barrier against bacteria and fungi. Also, the small size and large surface area [12,13,14,15] of AgNFs enhances their detection and response capability.

In this article, uniform silver nanofibers (AgNFs) are formed through a simple and cost-effective technique called electrospinning [16] with a controlled diameter [17] of nano-size in the fabrication of a sensor for sensing relative humidity, gas, and temperature with rapid response and recovery time because the relative humidity is not only important for human comfort but also important for different industries and technologies [18]. A practical humidity sensor must have the properties of excellent sensitivity, quick response and recovery time, chemical and thermal stability, and repeatability. Humidity sensors are of two types reported so far, one is the capacitive type [19] and the other is a resistive type [20]. A capacitive typesensor is used for polymers. A capacitive humidity sensor is noticed against the water molecules present in the air. The second type of humidity sensor is a resistive type whose resistance changes with water molecules. A resistive type humidity sensor is used for conductive materials like metals, copper, and silver. Similarly, permanent monitoring is needed for our surrounding environment to prevent the proliferation of toxic gases [21] such as NH_3_, which is a colorless and poisonous gas present in our atmosphere that has a great impact on various fields such as the industrial, chemical, and medical fields. NH_3_ at a higher concentration in the atmosphere can affect the human body and cause irritation of the eyes, throat infections, skin infections, and respiratory problems. For this reason, many sensors have been used so far like metal oxide gas sensors [22], catalytic ammonia detectors [23,24], optical ammonia [25], conductive polymer sensors [26,27], solid-state gas sensors [28], polyamaline base ammonia detection, optical gas analyzers, and reduced graphene oxide thin films, but quick response and recovery is the main problem associated with these sensors [29,30,31]. Also, the temperature is the main factor affecting the lifestyles of living things. It is the degree of hotness or coldness of a place measured in Kelvin, Fahrenheit, and Celsius. Temperature plays an important role in many fields, which is important for the mechanism that is used in medical drugs, liquids, laboratories, cosmetics products, goods, ICs fabrications, aerospace and defense, automotive, and many other fields. The accuracy of the temperature during the manufacturing is very important for quality control.

Herein, we report the synthesis of AgNPs through a chemical method, fabrication of AgNFs by an electrospining technique, and its sensing applications in relative humidity, gas and temperature sensing. It seems from the results that AgNFs exhibits superior sensing properties towards RH, NH_3_, and Temperature.

## 2. Experimental Work

### 2.1. Synthesis of Silver Nano Particles (AgNPs)

Before the synthesis of AgNFs, it is necessary to synthesize AgNPs [32]. For this purpose, a chemical method was used by taking 1 gm of silver acetate (AgCH_3_O_2_) and 2.5 mL of NH_3_ bought from Sigma Aldrich ChemieGmbh. Add the NH_3_ dropwise to the silver acetate and stir for a few minutes at 40 °C until the solution is completely dissolved (black color). Then add drop wise 0.2 mL formic acid (CH2O2) to the solution, and an exothermic reaction will take place according to the chemical formula and the solution turns into a grey color. Keep the solution for a night to settle down the particles at the bottom. The solution is then filtered through a 0.2 µm syringe filter. Finally, a transparent solution [33] is obtained called AgNPs.
AgCH_3_O_2_ + NH_3_→ [Ag (NH_3_)] CH_3_CO_2_(1)
[Ag (NH_3_)] CH_3_CO_2_ + CH_2_O_2_ → Ag + NH_3_ +HCO_2_ +HCO_2_(2)

### 2.2. Fabrication of Silver Nano Fibers (AgNFs)

For the synthesis of AgNFs, polyvinyl alcohol (PVA) solution (2 gm PVA and 20 mL distilled water) was added to 2 mL AgNPs, which was 10% of the total weight of the solution. The solution was stirred at 140 °Cat 350 r.p.m. Its viscosity was checked because at lower concentration electrospray occurs instead of electrospinning which creates a mixture of fibers and beads [34]. For the synthesis of AgNFsthroughthe electrospinning technique, the AgNPs solution was loaded in a medical/plastic syringe. The positive terminal of the power supply connected with the tip of the needle of the plastic syringe containing the solution. The negative terminal of the power supply was connected to the electrospinning collector. Aluminum foil was wrapped around the collector side. A power supply of 15 kva was used in the electrospinning unit. The distance between the tip of the needle and collector was kept from12 cm to 16 cm.A 12 vd.c motor was used in the electrospinning unit to push forward the plastic syringe for the continuous flow of solution. Also, the speed of the motor was kept at a minimum speed to prevent the solution from spoiling. Composite and uniform AgNFs were fabricated at the collector side [35].

## 3. Characterizations

Before the material can be used in different applications, it is necessary to characterize the physical and chemical properties, size, structure, and diameter of the AgNFs. For this purpose, different analytical techniques were used including Fourier transform infrared spectroscopy (FTIR), scanning electron microscopy (SEM), and X-ray diffraction (XRD). After analyzing, the AgNFs were then used for relative humidity, temperature and ammonia sensing. The flow chart for the whole research methodology used is shown in Figure 1.

### 3.1. Fourier Transform Infrared Spectroscopy (FTIR)

The FTIR of AgNFs was performed to check the bond formation using Perkin Elmer FTIR spectrometer machine model spectrum two, serial number 103385, scan speed 0.2, IR-Laser Wave number 11,750.00, Wavelength ranging from 4000 cm^−1^ to 450 cm^−1^, as shown in Figure 2.

The FTIR of AgNPs (black curve) shows an absorption peak at 899 cm^−1^, representing O-H aromatic compounds.After adding PVA solution to the AgNPs for the fabrication of silver nanofibers, the final FTIR was conducted (red curve). The peaks show the bond formation at 3283 cm^−1^ corresponding to O-H vibration, 2910 cm^−1^ shows C-H stretching, 1407 cm^−1^ shows C-H_2_ methylene groups, 1087 cm^−1^ shows C=O stretching, which shows a strong bond between 80%T to 40%T. Moreover, FTIR of PVA (green) in pure form was done and the peaks reported at wavelengths 3283 cm^−1^ and 1635 cm^−1^.

### 3.2. Scanning Electron Microscopy (SEM)

The structure and diameter of the AgNFs investigated through scanning electron microscopy (SEM), serial numbered JEOL 5910 SEM with a magnification range of 10X–300 K X and tungsten electronic beam energies in the range 1–40 keV. The diameter of the fibers was calculated through image J software. The fiber diameter investigated was uniform. As shown in Figure 3a, the diameters of nanofibers were 166 nm, Figure 3b 333 nm, Figure 3c 208 nm, and Figure 3d 285 nm.

### 3.3. X-RAY Diffraction (XRD)

Using a PANalytical-X’Pert³ powder X-ray diffractometer equipped with CuKα1radiation (=1.54056Å), installed in the materials research laboratory (MRL), Department of Physics, University of Peshawar, the XRDofAgNFs calcined at 350 °C was performed. The X-ray diffraction patterns were recorded at 2ϴ ranging from 10° to 70° with a scan speed of 1°/min and step size of 0.020. Furthermore, the values set for voltage and current during the operation were 45 kV and 40 mA, respectively. A singlephase of silver was reported, which confirmed the single composite nanofibers with no additional phases. The structure and peaks of the silver were matched using MATCH software. It was confirmed that the fiber structure was cubic. Peaks were found at angles 38°, 44°, and 64°, as shown in Figure 4.

### 3.4. Fabrication of Sensor

The interdigitated electrodes (IDE) used in many applications [36,37,38] were fabricated by depositing the AgNFs layer by electrospinning over it. After the deposition of AgNFs, the device was dried in an oven at 100 °C for 1 h. The distance between the comb fingers and the width of the comb fingers is 0.21 mm. The length, width, and thickness of the substrate are 14, 7, and 1 mm respectively (0.5 mm Alumina & 0.5 mm Copper layer). The schematic is given below. The entire process of sensor fabrication and schematic of the device is given in Figure 5a,b.

## 4. Results and Discussion

### 4.1. Relative Humidity

The AgNFsdeposited IDE was kept in a humidity chamber and resistance vs. RH (%) was investigated. Inside the chamber, the resistance of the IDE was calculated with respect to time and water molecules.It was observed that the resistance of IDE increases linearly with RH (%) [39], as shown in Figure 6.

The resistive response of the sensor was investigated at different frequencies, i.e., 100 Hz, 500 Hz, 1 kHz, 10 kHz, 100 kHz, and 1 MHz. It was observed that the response of the sensor is not very good at higher frequencies. The sensor shows excellent behavior up to 10 kHz. As the graph shows, relative humidity was measured from 30–90%RH, and the resistance was 4.3 kΩ from 100 Hz to10 kHz.

It is clear from the graph that measuring relative humidity from 30% to 90%, the sensor took 145 s to respond to maximum relative humidity and the resistance was 4.3 kΩ in the forward direction (curves starting from 0 s).The same experiment was repeated in the reverse direction (curves from 4.3 kΩ to 160 s), and the same data was investigated. Both the curves were found to be the same, which means that the sensor shows excellent response and recovery in both directions over equal intervals of time. It proved that the sensor is linear and has good performance.

### 4.2. Ammonia Sensing

The sensing device is also used to sense toxic gasses like NH_3_. The sensor shows a good response to NH_3_. For this purpose, different frequencies ranging from 100 Hz to 1 MHz were used. A resistance vsammonia concentration (ppm) graph was noted. It was found that at 100 Hz and 1 kHz, the resistance was 400 kΩ, and NH_3_ molecules were 40,000 ppm, which means that the sensor is a resistive type and shows excellent behavior towards NH_3_ molecules as shown in Figure 7.

It was also noted that the resistance is directly proportional to ppm. Resistance and ppm were co-linear. Resistance increases as ppm increases [40]. It was also confirmed that at low frequencies the response of the sensor is excellent compared to higher frequencies.

### 4.3. Silver Nanofibers Sensor vs. Temperature

It is obvious that most conducting materials change their resistance withtemperature [41].That is why specific resistance of a specific material is specified with standard temperatures. IDE coated with AgNFs was used as a temperature sensor by calculating its temperature with respect to resistance. The main idea is to confirm that the sensor is not affected by temperatures up to 69 °C. The resistance increases with temperaturesthat mean that the sensor has a positive temperature coefficient [42]. It was observed that at 5 MΩ, the temperature was 69 °C, as shown in Figure 8.

### 4.4. Response and Recovery Time

For an excellent sensing device, it is important that it exhibits a rapid response and recovers to its initial state in a short period of time. To check the performance of the designed sensing device, response and recovery measurements of the device have been performed, as shown in Figure 9. From the figure, it was observed that the device shows a maximum resistive sensitivity at lower frequencies due to skin or surface effect, because the gas molecules directly interact with the surface of the material and at lower frequency skin effect is dominant over a bulk material. Also, the device sensitivity was observed at different concentrations of NH_3_ in parts per million (ppm). When the device is exposed to NH_3_ in a chamber, the gas molecules interact with the surface of the sensing material and thus increase the resistance of the device from 25 kΩ to 450 kΩ at 100 Hz and 1 kHz, respectively. The response time is the time from the initial state to the maximum saturation state of the device, while the recovery time is the period from maximum saturation back to the initial state. The maximum response time of the device observed was 27 s (from 49 to 71 s on the x-axis time scale) at 20,000 ppm (blue strip) while the maximum recovery time of the device was observed at 33 s at 10,000, ppm highlighted in yellow in Figure 9.

## 5. Conclusions

In this research, the structural morphology, chemical, and physical properties of silver nanofibers (AgNFs) have been studied. Different techniques have been performed for characterization, and the results led to the following conclusions.

AgNPs have been synthesized using ammonia, formic acid, and silver acetate through a chemical method. The AgNPs obtained were transparent in color. The AgNPs solution (2 mL) was then added to 20 mL of PVA solution to synthesize AgNFs through the electrospinning technique. The synthesized AgNFs were then kept in the oven to dry at 90 °C. The dried fibers were then kept in the furnace at 350 °C for 2 h to calcine and to reduce the diameter of the fibers.

SEM images show the diameter of the nanofibers in nanometers. The FTIR of nanofibers shows the chemical bond formation at wavelengths, 3283 cm^−1^, 2910 cm^−1^, 1635 cm^−1^, 1407 cm^−1^, 1087 cm^−1^, and 899 cm^−1^. The calcined nanofibers were then characterized through XRD. Peaks reported at 38°, 44°, and at 64° with intensities of 740, 200, and 170, respectively. The AgNFs were found to be cubic in shape.

The AgNFs deposited IDE device was used for sensing relative humidity, and it was found to be of a resistive type. At 4.3 kΩ, the relative humidity was 90%. The response and recovery time of the sensor was noted, which was 145 s. It was reported that at lower frequencies the device shows excellent behavior i.e., at 100 Hz and 10 kHz, the relative humidity was 90%.

The sensor was then used for temperature sensing and it was found that the sensor has a positive temperature coefficient. It was observed that at 0.5 MΩ the temperature was 45 °C, at 1.5 MΩ the temperature was 55 °C, and at 5 MΩ the temperature was 69 °C, which means that resistance increases with temperature.

Similarly, the sensor was used for gas sensing applications for ammonia. It was found that at 100 Hz and 1 kHz, the resistances measured were 400 kΩ, 275 kΩ, and 150 kΩ for NH_3_ concentrations of 40,000 ppm, 20,000 ppm, and 10,000 ppm, respectively. The sensor starts sensing after 13 s, andtakes 27 s (rise time) to respond to ammonia molecules to their maximum value and after releasing ammonia molecules it returns to its original value after 33 s (fall time), which shows slow recovery at a lower concentration of ammonia.

Finally: it was concluded that the device has a linear and repetitive response, is cost-effective, and is easy to fabricate. Also, it seems from the results that AgNFs exhibits superior sensing properties towards RH, NH_3_, and temperature.

## Figures and Tables

**Figure 1 micromachines-12-00682-f001:**
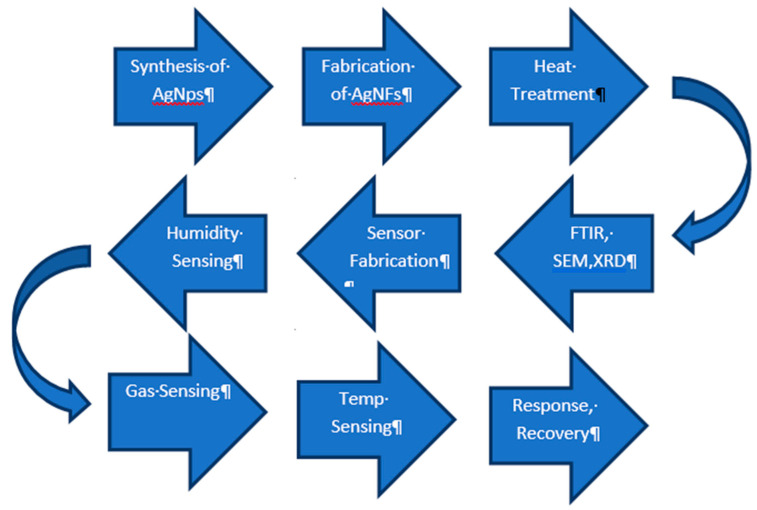
Flow chart for the research methodology, characterization, fabrication, and sensing.

**Figure 2 micromachines-12-00682-f002:**
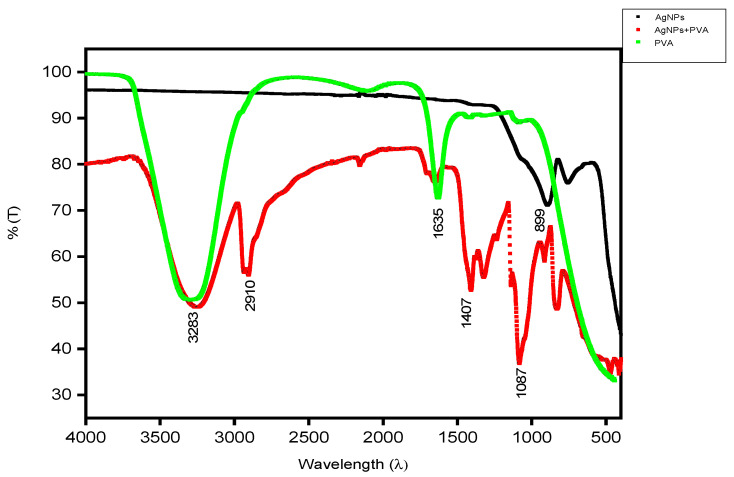
FTIR of AgNPs + PVA (red), PVA (green), and AgNPs (black).

**Figure 3 micromachines-12-00682-f003:**
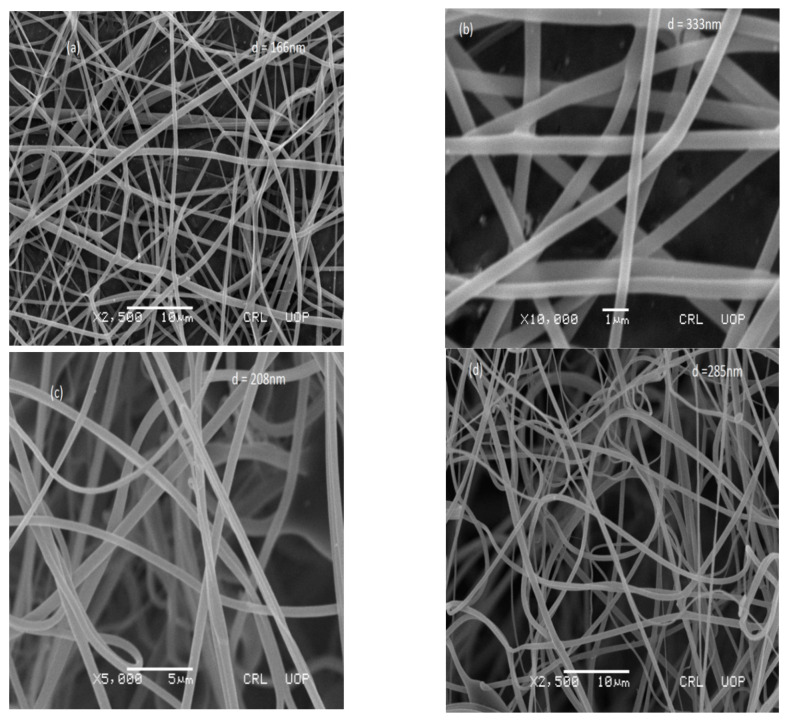
(**a**–**d**): SEM images with diameters in nanometers.

**Figure 4 micromachines-12-00682-f004:**
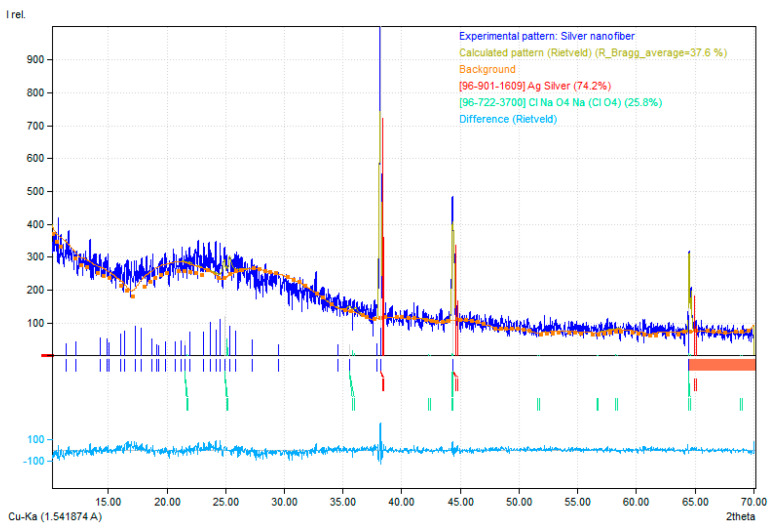
X-ray Diffractionspectrumof Silver nanofibers.

**Figure 5 micromachines-12-00682-f005:**
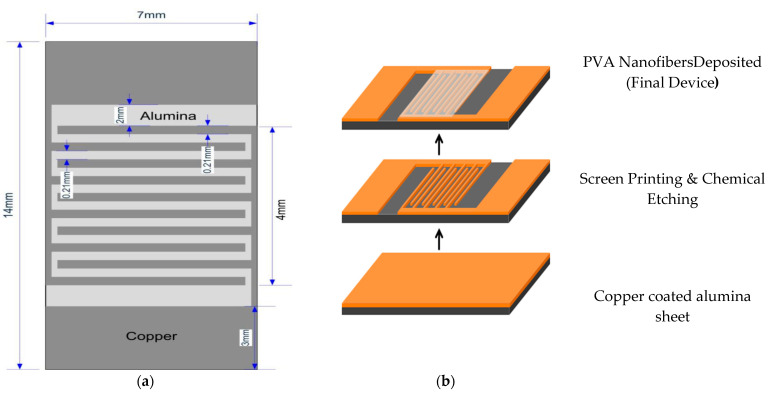
(**a**) Dimention (**b**) Printed diagrma ofinterdigitated electrode.

**Figure 6 micromachines-12-00682-f006:**
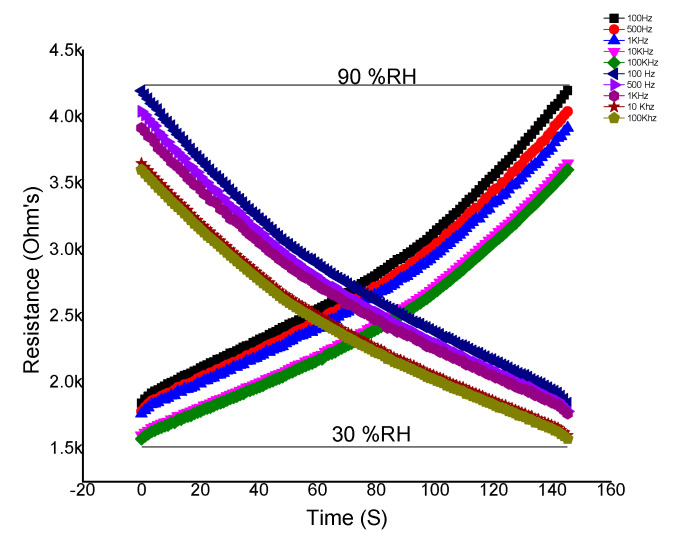
Relativehumidity (30–90%), frequency, and time vs. resistance.

**Figure 7 micromachines-12-00682-f007:**
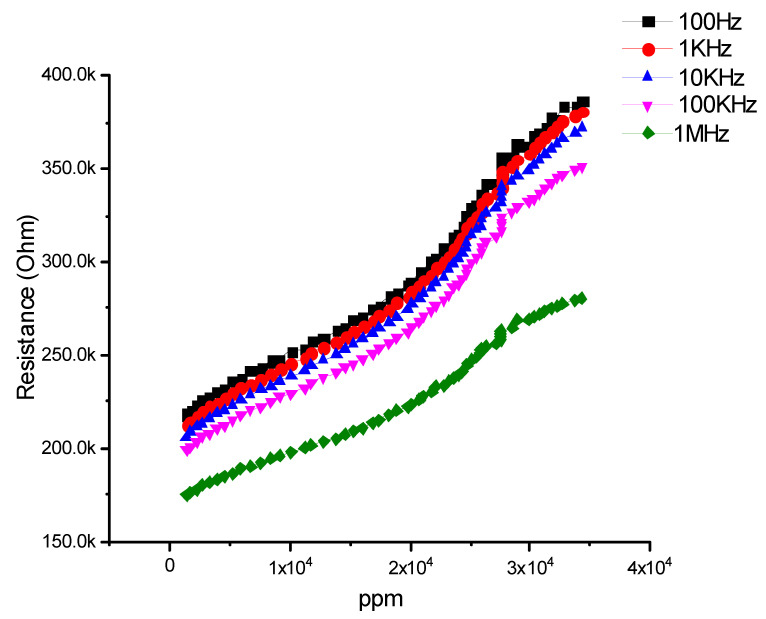
Resistance (ohm) vs. concentration of ammonia (ppm).

**Figure 8 micromachines-12-00682-f008:**
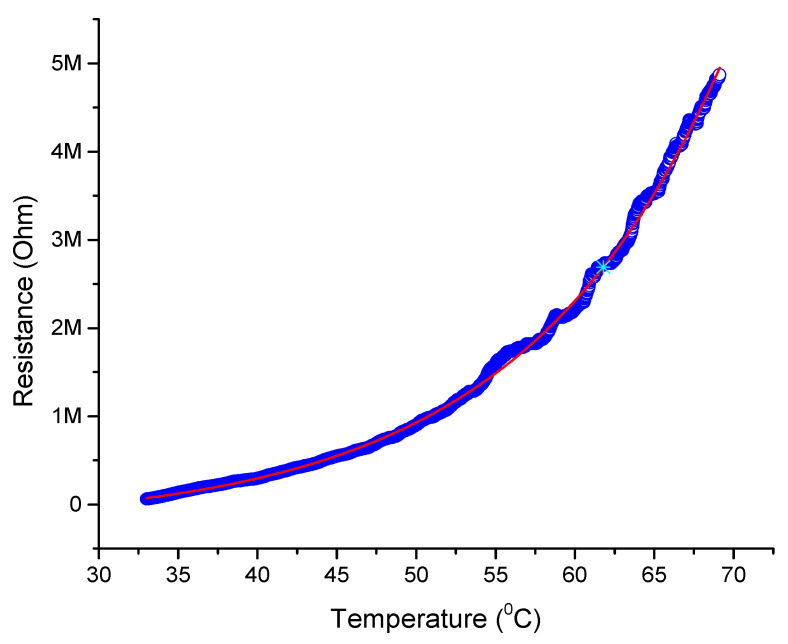
Resistance of the fabricated silver nanofibers sensor vs. temperature.

**Figure 9 micromachines-12-00682-f009:**
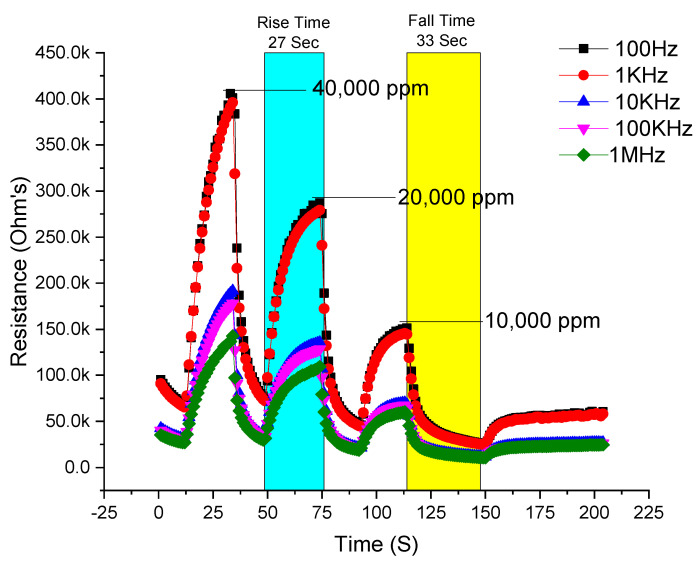
Response (rise of the curve) and recovery (fall of the curve) of the ammonia sensor.

## Data Availability

Not applicable.

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
