# Peer review of "Synthesis, Characterization, and Applications of Silver Nano Fibers in Humidity, Ammonia, and Temperature Sensing"

_micromachines, 2021, doi:10.3390/mi12060682_

Round 1

Reviewer 1 Report

The paper deals with the idea of a novel method for the fabrication of Silver Nano Fibers (AgNFs) which is environmentally friendly 13 and can be easily deployed for large-scale production; also the proposed technique is easy for device 14 fabrication in different applications. The paper is basically well written and offers interesting findings. Several points which must be improved: 

  1. Avoid lumping many references without providing sufficient description. Please provide a short description or descriptor for each used reference, therefore, the readers can understand the contents of the used reference.
  2. The novelty should be justified and explained clearly further. Currently, it is difficult to measure the novelty of the paper, including the significant difference with the previous studies and existing literature. 
  3. The units must be written correctly, and provide a space between the value and unit. 
  4. Further detailed discussion with additional references are required for discussion section. For example, in section 4.3, additional description about any physical phenomenon is important, as well as theoretical explanation.

Reviewer 2 Report

Raw 5, please correct.

Raw 121, Figure 2, please add units at Wavelength

Raw 206 “….. within no time”, please reformulate.

In the discussion section, you can also refer to electrical conductivity dependence on the silver amount, from a recent study (DOI: 10.1109/jsen.2019.2935232) or some other literature studies by doing a comparison with your present results.

Reviewer 3 Report

Generally speaking, the manuscript is well written, the material is judiciously divided and organized and correct from scientific point of view. Some changes are, however, necessary. For these reasons I can recommend the acceptance of this paper after some corrections presented in the attached file.

Round 2

Reviewer 1 Report

The authors failed to correct the paper. 
Many units are still written incorrectly, as well as the space is not provided between the value and unit throughout the manuscript. In addition, the symbol to represent the degree is incorrect (try to find the appropriate symbol, and don't use superscripted zero). 

Although the unit seems to be simple, it reflects the quality and understanding of the authors. I think you need to investigate more deeply how to write unit correctly.

Reviewer 3 Report

No comments